# Enhancing Fairness in Meta-learned User Modeling via Adaptive Sampling

## ABSTRACT

Meta-learning has been widely employed to tackle the cold-start problem in user modeling. The core idea is to learn the globally shared meta-initialization parameters for all users and rapidly adapt them to user-specific local parameters. Similar to a guidebook for a new traveler, meta-learning significantly affects decision-making for new users in crucial scenarios, such as career recommendations. Consequently, the issue of fairness in meta-learning has gained paramount importance. Several methods have been proposed to mitigate unfairness in meta-learning and have shown promising results. However, a fundamental question remains unexplored: What is the critical factor leading to unfairness in meta-learned user modeling? Through the theoretical analysis that integrates the meta-learning paradigm with group fairness metrics, we identify group proportion imbalance as a critical factor. Subsequently, in order to mitigate the impact of this factor, we introduce a novel Fairness-aware Adaptive Sampling framework for meTa-learning, abbreviated as FAST. Its core concept involves adaptively adjusting the sampling distribution for different user groups during the interleaved training process of meta-learning. Furthermore, we provide theoretical guarantees demonstrating the convergence of FAST, showcasing its potential to effectively eliminate unfairness. Finally, empirical experiments conducted on three datasets reveal that FAST effectively enhances fairness while maintaining high accuracy.

## KEYWORDS

meta-learning, fairness, user modeling, adaptive sampling

## 1 INTRODUCTION

User modeling aims to infer latent characteristics by analyzing users' behavioral information, such as capability and preference, and is widely applied in numerous web applications (e.g., recommender system) [46]. One common challenge in user modeling is the cold-start problem [13], which arises when interactions are severely limited for new users. In recent years, meta-learning methods [2, 17, 30, 35] have garnered widespread adoption as a solution to this issue, which facilitate rapid learning from limited data while remaining computationally efficient. The core idea behind meta-learning is "learning-to-learn" [11], i.e., learning globally shared meta-initialization parameters for all users and then swiftly adapting them into local parameters specific to each user. The training approach involves an interleaved training procedure, as illustrated in Figure 1. It contains 1) an inner loop that local updates the user-specific parameter $\theta_i$, initialized by meta-parameters $\theta$; 2) an outer loop that global updates the meta-parameter $\theta$.

As meta-learning, much like a guidebook for a new traveler, significantly influences the decision-making of fresh users in crucial scenarios, such as career recommendations [16] and college admissions [15], the fairness problem has attracted significant attention from a broad audience. It requires user groups divided by

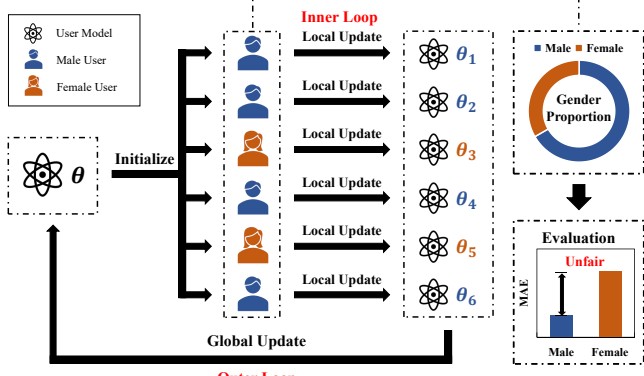

**Figure 1: Illustration of unfairness in meta-learning for user modeling. In the outer loop optimization, group proportion imbalance results in unfairness between different groups.**

sensitive attributes (e.g., gender, race) should be treated similarly. Regrettably, meta-learning methods have exhibited susceptibility to unfairness [34, 41, 42, 45], prompting the development of numerous fairness-aware meta-learning approaches, such as regularization-based methods [26, 43], constraint-based methods [41, 42], and adversarial-based methods [34].

While these methods have demonstrated promising results, a fundamental question remains unexplored: ***What is the critical factor leading to unfairness in meta-learned user modeling?*** Understanding this question not only enhances our comprehension of the meta-learning paradigm but also enables us to make targeted improvements for superior fairness outcomes. To address this inquiry, we conduct an in-depth theoretical analysis by combining the meta-learning paradigm with fairness metrics. Specifically, we establish a correlation between the lower bounds of the unfair degree in meta-learning and the level of imbalance in group proportions. Our theoretical findings indicate that during the outer loop optimization, as the group proportions become more imbalanced, the lower bound increases, making unfairness more probable. This theoretical contribution underscores that a critical determinant of fairness in meta-learning is the ***group proportion imbalance*** during the outer loop (as shown in Figure 1).

Having gained insight into this factor contributing to unfairness in meta-learning, a second question arises: ***How can we mitigate the influence of the factor to enhance fairness in meta-learned user modeling?*** In response to this question, we introduce a novel framework, called Fairness-aware Adaptive Sampling for meTa-learning, abbreviated as FAST. Our core concept involves the adaptive adjustment of group-level sampling distribution during the training process. Specifically, within the inner loop of meta-learning, we introduce a Group-level Binary Cross Entropy (G-BCE) loss to continuously monitor the model's fairness status at each step. Then, in the outer loop, by leveraging the fairness signals gathered

during the inner loop, we propose an adaptive update strategy to optimize the group sampling distribution. This effectively addresses the issue of group proportion imbalance, thus promoting fairness. Notably, we offer theoretical evidence showcasing that FAST can make the sampling probabilities of each group converge to the optimal values that achieve the minimal unfair degree. This suggests that from a theoretical standpoint, our FAST has the potential to completely eliminate unfairness.

Finally, we apply FAST to some classic meta-learning models for user modeling. The corresponding code is readily accessible at https://anonymous.4open.science/r/FAST-7516/. Empirical experiments conducted on three datasets reveal that FAST effectively enhances fairness while maintaining high accuracy. Our key contributions in this work can be summarized as follows:

- **Understanding Fairness.** We theoretically identify group proportion imbalance as a crucial factor contributing to unfairness in meta-learned user modeling.
- **Improving Fairness.** We introduce a general framework, FAST, to facilitate fairness optimization for meta-learning. Additionally, we offer theoretical guarantees demonstrating FAST's potential to completely eliminate unfairness.
- **Fairness Evaluations.** Empirical experiments demonstrate that FAST is highly effective in enhancing fairness without compromising accuracy.

## 2 RELATED WORKS

### 2.1 Meta learning for User Modeling

The cold-start problem represents a prevalent challenge in user modeling, wherein the interactions are significantly constrained for new users [13, 23, 37, 44]. In recent years, meta-learning has gained substantial traction in effectively tackling the cold-start problem in user modeling, yielding remarkable success [2, 11, 17, 27]. This innovative approach empowers models to swiftly grasp new tasks with limited labeled data by leveraging prior knowledge gleaned from preceding tasks. For example, Lee et al. [17] introduced MELU, a method that harnessed MAML [11], to ascertain the initial neural network weights for cold-start users. Building upon the MELU concept, Bharadhwaj et al. [2] adopted a similar approach in constructing their recommender model. Furthermore, they employed a more versatile meta-update strategy to refine the model parameters. Wang et al. [32] proposed a preference-adaptive meta-learning approach, which enhanced existing meta-learning frameworks by bolstering their generalization capacity. Ye et al. [40] advocated for a personalized adaptive meta-learning approach that tackled the issue of non-uniformly distributed data. Wei et al. [34] presented CLOVER, a comprehensive meta-learning framework designed to ensure the fairness of meta-learned recommendation models. In addressing the cold start problem in cognitive diagnosis [31], a classical user modeling task, Bi et al. [3] proposed a comprehensive Bayesian meta-learned framework. Nonetheless, the origins of unfairness within the meta-learned user modeling remain relatively unexplored. In this study, we delve into alleviating the fairness concerns inherent in meta-learning for user modeling.

### 2.2 Fairness in User Modeling

As user modeling is widely applied in various scenarios, such as recommender systems [14, 33], cognitive diagnosis [24]. The fairness issue has emerged as a crucial concern. There are numerous perspectives to study fairness [8, 9, 18, 20, 38]. For instance, Yao et al. [39] introduced novel metrics for collaborative filtering-based recommendations to quantify the disparity in prediction behavior between disadvantaged and advantaged users. Wu et al. [36] approached fairness from a graph-based perspective, simultaneously proposing an adversarial learning framework to enhance fairness. Zhu et al. [45] explored the issue of fairness in new item activity and devised a trainable post-processing framework to improve fairness. Li et al. [21] investigated counterfactual fairness in recommendation systems and endeavored to minimize the mutual information between user embeddings and sensitive attributes to meet the requirements of counterfactual fairness. Li et al. [19] proposed a user-oriented group fairness definition, arguing that algorithms should deliver an equal level of utility performance for different user groups. This fairness definition has gained wide acceptance [10, 12, 20, 34]. In this paper, we also adhere to this definition.

Regarding the investigation into the causes of unfairness for user modeling, many researchers have pointed out group imbalance as a significant factor while simultaneously proposing solutions. For example, Chen et al. [5] addressed the issue of mitigating imbalanced training data through a data augmentation perspective. Chen et al. [6] proposed a fairly adaptive negative sampling approach to improve item group fairness for pairwise algorithms. Different from these approaches, to the best of our knowledge, we are the first to theoretically identify group proportion imbalance as a critical factor in meta-learning. Simultaneously, we have elucidated that this imbalance leads to unfairness in the outer loop of the meta-learning process and have provided corresponding solutions with strong theoretical guarantees.

### 2.3 Fairness in Meta-learning

With the evolution of meta-learning, an increasing number of studies have taken up the challenge of addressing fairness concerns in this domain. These approaches can be broadly categorized into three primary types based on their optimization strategies: regularization-based methods [26, 43], constraint-based methods [41, 42], and adversarial-based methods [34]. Slack et al. [26] and Zhao et al. [43] introduced the incorporation of group regularization terms into the meta-learning optimization process. Furthermore, Zhao et al. [42] pioneered an innovative framework for fair meta-learning that integrates task-specific group soft fairness constraints. Building upon their prior work, Zhao et al. [41] extended their research to confront the intricate issue of online fairness-aware meta-learning, which deals with non-stationary task distributions by integrating long-term fairness constraints. Wei et al. [34] introduced an adversarial-based approach designed to simultaneously achieve three types of user-oriented fairness in meta-learning. While previous studies have demonstrated promising results, they have tended to overlook a fundamental question: What is the root cause of unfairness in meta-learned user modeling? In this paper, we theoretically identify group proportion imbalance as a crucial factor and present an adaptive sampling strategy to mitigate the impact of this factor.

## 3 PRELIMINARIES

In this section, we introduce the cold-start problem definition and the fairness considerations in user modeling.

### 3.1 Problem Definition

Suppose there are $M$ users, represented as $U = \{u_1, u_2, ..., u_M\}$. Each user $u_i$ has a limited set of rated items, denoted as $I_i^s$. The corresponding rating between user $u_i$ and each item $v^s \in I_i^s$ is denoted as $r_{iv^s}$. The primary objective of the system is to predict the rating $r_{iv^q}$ by user $u_i$ for a new query item $v^q \in I_i^q$, where $I_i^q$ represents the set of items that need to be predicted.

For each user, there are two datasets available: the support set for fine-tuning and the query set for testing. The support set for user $u_i$ is denoted as $D_i^s = \{u_i, v^s, r_{iv^s}\}, v^s \in I_i^s$, and the query set is denoted as $D_i^q = \{u_i, v^q, r_{iv^q}\}, v^q \in I_i^q$. Following previous works [17, 34, 35], we consider the fast adaptation for each user $u_i$ as a task $t_i$:

$$t_i : f(\theta, D_i^s) \rightarrow \theta_i, \tag{1}$$

where $\theta$ is the meta-parameters, and $\theta_i$ is the personalized parameter fine-tuned on $D_i^s$ for each user. The focus of the cold-start user modeling system is on new users who arrive after the training stage. Let $U^c$ represent the set of fresh users that will enter the system. Our goal is to learn proper $\theta$ on existing users that generalize well to new users $U^c$, achieving efficient fast adaptation on their tasks.

### 3.2 User-oriented Group Fairness

In this paper, we focus on user-oriented group fairness, which requires that user groups divided by sensitive attributes (gender, race) should be treated similarly. For the sake of analytical convenience, we focus on the case where the sensitive attribute is binary, which can be easily extended to multiple values. The user group can be divided into two groups based on sensitive attributes, denoted as $G_0$ and $G_1$, where $U = G_0 \cup G_1, G_0 \cap G_1 = \varnothing$. We denote the number of samples in each group as $m_0$ and $m_1$, respectively. Without loss of generality, we assume $m_1 > m_0$. Adhering to the widely adopted user modeling fairness definition, a fair user modeling algorithm should provide the same level of utility performance for various user groups [19]. Subsequently, user-oriented group fairness in user modeling (GF) is defined as follows, lower GF represents better fairness performance:

DEFINITION 1 (USER-ORIENTED GROUP FAIRNESS).

$$GF = \left| \frac{1}{m_0} \sum_{u_i \in G_0} \mathcal{M}(u_i) - \frac{1}{m_1} \sum_{u_i \in G_1} \mathcal{M}(u_i) \right|, \tag{2}$$

where $\mathcal{M}$ represents a metric for evaluating utility performance, such as MAE or MSE score, and $\mathcal{M}(u_i)$ to denote the utility performance for user $u_i$.

After introducing the user-oriented group fairness definition, our objective extends beyond merely generalizing well to new users. We should also strive to meet fairness requirements, aiming to minimize the GF.

## 4 UNDERSTANDING FAIRNESS IN META-LEARNING

In this section, we begin with introducing the classic meta-learning paradigm. Subsequently, we delve into a theoretical exploration of the underlying factors that lead to unfairness in meta-learning for user modeling.

### 4.1 Meta-learning for User Modeling

The core concept underlying meta-learning for user modeling is learning to learn, i.e., learning to solve the rapid adaptation task $(t_i)$ for new users. To clarify this concept, we adhere to a widely recognized framework [2, 17, 34]. In order to learn the parameters for a new task (user), the primary objective of the meta-model is to establish a promising initialization through learning from a variety of similar tasks. Subsequently, leveraging the learned parameters, the meta-model can be fine-tuned on the new task with limited interactions. The workflow for meta-learning in user modeling is visually depicted in Figure 2. The training methodology involves an interleaved training procedure, including:

**Inner loop.** In the inner loop, for each user $u_i$, the framework initializes $\theta_i$ with the most recent meta-parameter $\theta$, and then proceeds to fine-tune $\theta_i$ based on the user's existing training data $D_i^s$ as follows:

$$\theta_i \leftarrow \theta_i - \alpha \nabla_{\theta_i} L(f_{\theta_i}, D_i^s), \tag{3}$$

where $\alpha$ is the learning rate of parameter update, $Loss(f_{\theta_i}, D_i^s)$ denotes the prediction loss (e.g., cross-entropy loss ) on data $D_i^s$. $f_{\theta_i}$ suggests the loss is parametrized by parameter $\theta_i$.

**Outer loop.** In the outer loop, the meta-parameter $\theta$ is updated by summing up all user $u_i$'s specific loss $(Loss(f_{\theta_i}, D_i^q))$ on query data set $(D_i^q)$, and then minimizing them to provide a promising initialization for each user. More precisely, during each training step, the meta-parameter $\theta$ is updated as follows:

$$\theta \leftarrow \theta - \beta \nabla_\theta \sum_{u_i \in B} L(f_{\theta_i}, D_i^q), \tag{4}$$

where $B$ is a set of users involved in the batch, $\beta$ is the learning rate of meta-parameters. Note that the query set is in the training set.

After detailing the training process of the meta-learning, we will proceed to explain the testing process. To evaluate a new user $u^c \in U^c$, the framework will begin by initializing the user model parameters using the meta-model $\theta$. Subsequently, the user model undergoes fine-tuning with the user's observed interaction data $D_{u^c}^s$. Finally, the fine-tuned model will be employed to make decisions.

### 4.2 Fairness Understanding

Meta-learning paradigms have already been extensively studied and they significantly impact the decision-making processes of new users in critical scenarios. Consequently, it is important to investigate fairness issues in meta-learning. However, a fundamental question remains unexplored: What is the critical factor leading to unfairness in meta-learned user modeling? In this section, we attempt to analyze this question by integrating the meta-learning paradigm with relevant fairness metrics.

We represent the average loss of group $G_0$ and group $G_1$ as $\mathcal{L}_0$ and $\mathcal{L}_1$, where $\mathcal{L}_0 = \frac{1}{m_0} \sum_{u_i \in G_0} L(f_{\theta_i}, D_i^q)$ and $\mathcal{L}_1$ is calculated

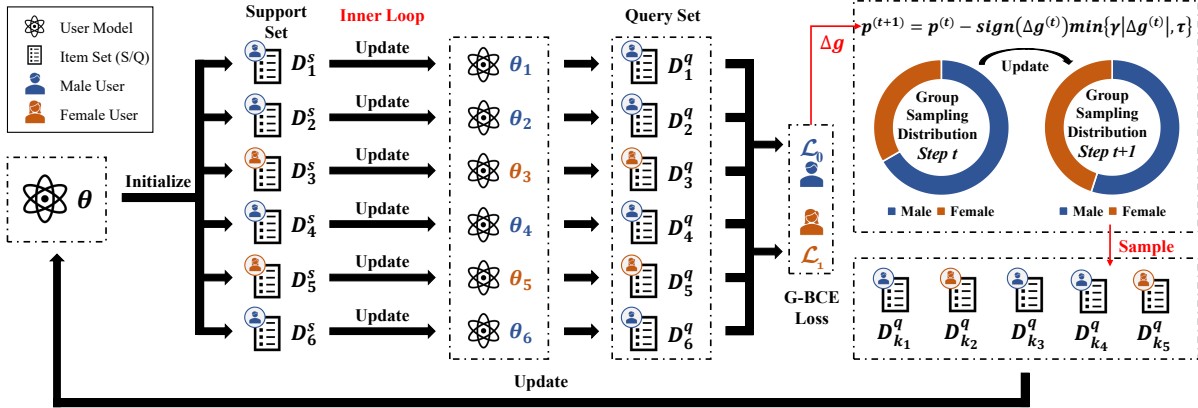

**Figure 2: The overall framework of FAST. Compared to traditional meta-learning, we introduce an adaptive sampling strategy during the outer loop. This strategy can adaptively adjust the sampling probabilities for each group, mitigating the impact of group proportion imbalance.**

in a similar manner. Following the classical group fairness definition introduced in section 3.2 and previous works [6, 40], the gap between these average losses, $\mathcal{L}_1 - \mathcal{L}_0$, can serve as a measure of the degree of unfairness. Next, we will explore the factors that influence $\mathcal{L}_1 - \mathcal{L}_0$ through the following theorem.

THEOREM 1. *Given the user group $U$, comprising $M$ users, which can be divided into two subgroups, $G_0$ and $G_1$, based on binary sensitive attributes. Let the number of users in each group be represented as $m_0$ and $m_1$, respectively. Without loss of generality, let us assume $m_1 > m_0$. We further assume the actual preference of user $u_i$ is $x_i$, and the actual preferences of users in group $G_0$ and group $G_0$ are drawn from two normal distribution, i.e., $\mathcal{N}(-\eta, \sigma^2)$ and $\mathcal{N}(\eta, \sigma^2)$, respectively[1]. Then we have: with probability at least $1 - \frac{A}{\frac{m_1-m_0}{2M}}$, the loss gap $\mathcal{L}_1 - \mathcal{L}_0$ satisfies:*

$$\mathcal{L}_1 - \mathcal{L}_0 > \frac{m_1 - m_0}{2M}\eta^2 C, \qquad (5)$$

*where $C$ is a constant related to the learning rate $\alpha$, and $A = C^2[2(1 - \frac{1}{M})^2\sigma^4(\frac{1}{m_0} + \frac{1}{m_1}) + 16(1 - \frac{1}{M})\eta^2(\frac{m_0^3+m_1^3}{m_0 m_1 M^2})]$.*

The detailed proof is provided in Appendix A.1. According to Theorem 1, we can find that the lower bounds of $\mathcal{L}_1 - \mathcal{L}_0$ (i.e., $\frac{m_1-m_0}{2M}\eta^2 C$) is related to $m_1 - m_0$. To be specific, as the larger $m_1 - m_0$ (the group proportions become more imbalanced), the larger lower bound, making unfairness more likely to occur. This theoretical contribution underscores that a critical determinant of fairness in meta-learning is the group proportion imbalance.

To better understand Theorem 1, we offer an intuitive illustration here. In the outer loop optimization of meta-learning, all user records are utilized to update the meta-parameter $\theta$. Intuitively, the training process will prioritize users in major groups since the losses of these users occupy a larger proportion of the overall loss. Consequently, the learned $\theta$ will be closer to the centroid of major groups, which results in a disparity of average losses between major and minor groups.

[1]The choice of $-\eta$, $\eta$ as values is made for the sake of analytical convenience and does not impose any strict limitation. In reality, these values can be any two distinct numbers since we have the flexibility to adjust our coordinates to achieve symmetry.

## 5 THE PROPOSED FRAMEWORK

After recognizing how imbalanced group proportions can contribute to unfairness in meta-learning, this section introduces our novel framework, FAST, designed to mitigate the impact of this factor. The subsequent discussion will begin with an overview of FAST, followed by a detailed description of each of its components and an illustration of the designed optimization algorithm. Finally, we present the theoretical guarantees for FAST.

### 5.1 An Overview of the Proposed Framework

In the traditional meta-learning paradigm, within the outer loop, all user records are utilized to update the meta-parameters, with limited consideration for the imbalanced group proportions. This oversight can lead to noticeable fairness issues within the model. To overcome these limitations without making explicit alterations to the model architecture or training data, we introduce an adaptive sampling framework known as FAST, as depicted in Figure 2. The core concept of FAST lies in adaptively adjusting the sampling probability for each group during the outer loop. Specifically, the FAST method consists of two crucial components.

- **Group Fairness Perception**. This component is specifically designed to detect unfairness among groups during the inner loop of the meta-learning process, offering pertinent signals to guide subsequent sampling adjustments.
- **Adaptive Update of Group Sampling Distribution**. Building upon the unfairness signals gathered in the initial stage, this component adapts the sampling probability for each group in the outer loop, with the aim of attaining equalized performance across all groups.

### 5.2 Group Fairness Perception

To achieve fair training for meta-learning from the perspective of adaptive sampling, we first need to perceive the group-wise performance disparity in the inner loop, which provides guidance for adjusting the group-level sampling distribution in the outer loop. However, the fairness evaluation metric mentioned above cannot be used here directly since it is non-differentiable. Inspired by [6], we

adopt a Group-level Binary CrossEntropy (G-BCE) loss as a proxy to measure each group's performance. It calculates the group average classification loss as an approximation of group performance. The details of G-BCE loss are as follows:

$$\mathcal{L}_0 = \frac{1}{m_0} \sum_{u_i \in G_0} Loss(f_{\theta_i}, D_i^q), \ \mathcal{L}_1 = \frac{1}{m_1} \sum_{u_i \in G_1} Loss(f_{\theta_i}, D_i^q), \quad (6)$$

where $D_i^q$ represents the query set used to evaluate the performance of user $u_i$ during the inner loop. The performance gap between two groups, $G_1$ and $G_0$, can subsequently be expressed as follows:

$$\Delta g = \mathcal{L}_1 - \mathcal{L}_0. \quad (7)$$

The inequality $\Delta g > 0$ indicates that Group $G_1$ achieves lower performance. In other words, the meta-learning process has learned meta-parameters that generalize better to $G_0$ compared with $G_1$, resulting in unfair treatment of Group $G_1$. Furthermore, a larger value of $\Delta g$ signifies a higher degree of unfairness, necessitating a more substantial adjustment. Next, we will concentrate on the adjustment of group sampling probabilities during the outer loop to acquire meta-parameters that exhibit greater fairness.

## 5.3 Adaptive Update of Group Sampling Distribution

After detecting the direction and extent of unfairness among groups, we consistently modify the sampling probabilities of each group to mitigate performance disparities. We define the probability of sampling user $u_i$ belonging to Group $G_0$ as $p$ and belonging to Group $G_1$ as $1 - p$. In the following section, we will demonstrate the update mechanism for $p$. Intuitively, when $\Delta g > 0$, it indicates that Group $G_1$ achieves lower performance. To narrow the performance gap between $G_1$ and $G_0$ to ensure fairness, we should increase the sampling probability of $G_1$ samples while reducing the sampling probability of $G_0$ samples. Conversely, when $\Delta g < 0$, we should increase the sampling probability of $G_0$ samples. Based on these considerations, we design the following update strategy:

$$p^{(t+1)} = p^{(t)} - \gamma \Delta g^{(t)}, \quad p^{(0)} = 0.5, \quad (8)$$

where $\Delta g^{(t)}$ represents the performance gap between $G_1$ and $G_0$ in step t, $p^{(t)}$ and $p^{(t+1)}$ represent the sampling probabilities for $G_0$ at steps t and t+1, respectively. In the initial stages of optimization, due to a lack of information regarding the fairness between the two groups, we opt to initialize the sampling probabilities for both groups equally, setting them both to 0.5. Furthermore, $\gamma$ is a hyperparameter controlling the rate of change, and its influence will be studied in the experimental section.

The variation in the sampling process probabilities depends on $\Delta g$. However, during the training process, it is possible for the value of $\Delta g$ to change significantly due to unstable factors. This can lead to a substantial fluctuation in sampling probabilities, resulting in an unstable outcome in terms of fair training effectiveness. To mitigate this phenomenon, we introduce an upper threshold ($\tau$) for the changing sampling probabilities. Specifically, when the value of the changed sampling probability (i.e., $\gamma|\Delta g|$), exceeds $\tau$, we opt to replace it with $\tau$. The specific formula is as follows:

$$p^{(t+1)} = p^{(t)} - sign(\Delta g^{(t)})min\{\gamma|\Delta g^{(t)}|, \tau\}, \quad (9)$$

where $sign(\Delta g^{(t)})$ represents the sign of $\Delta g^{(t)}$, and we will investigate the impact of the hyperparameter $\tau$ in our experiments.

---

**Algorithm 1** The FAST framework

---

**Require:** $\alpha$, $\beta$: learning rate hyper-parameters; $\gamma$, $\tau$: the hyper-parameters of adaptive sampling;

1: Randomly initialize meta-parameters $\theta$ and set group sampling distribution $p_{z_0}^{(0)} = p_{z_1}^{(0)} = 0.5$;
2: **while** not converge **do**
3:    Sample batch of users $B$;
4:    **for** user $u_i$ in $B$ **do**
5:      set $\theta_i = \theta$;
6:      Inner loop update:
7:        $\theta_i \leftarrow \theta_i - \alpha \nabla_{\theta_i} L(f_{\theta_i}, D_i^s)$;
8:    **end for**
9:    Adaptive sampling:
10:      Calculate G-BCE loss according to Eq. (7);
11:      Update group's sampling probabilities based on Eq. (9);
12:      Sample the users $B'$ based on the sampling probabilities;
13:    Outer loop update:
14:      $\theta \leftarrow \theta - \beta \nabla_{\theta_i} \sum_{u_i \in B'} L(f_{\theta_i}, D_i^q)$;
15: **end while**
16: /* Testing on the new users $U^c$ */
17: **for** user $u^c$ in $U^c$ **do**
18:    Finetune on test data $D_c^{test}$ of $u^c$:
19:      $\theta_c \leftarrow \theta - \alpha \nabla_{\theta_c} L(f_{\theta_c}, D_c^{test})$;
20:    User rating prediction based on $\theta_c$;
21: **end for**

---

## 5.4 Optimization for FAST

In the preceding sections, we have introduced the training paradigm of traditional meta-learning, identified factors contributing to its unfairness, and outlined how an adaptive sampling strategy can be employed to mitigate these issues. In this section, we will delve into the integration of this sampling strategy into the training paradigm of traditional meta-learning to enhance its fairness. The overall training algorithm is depicted in Algorithm 1.

To begin, we will adhere to the conventional meta-learning paradigm and fine-tune each user within the inner loop. After obtaining the loss for each individual in the query set, we will calculate the G-BCE loss for each user group according to Eq. (7). Subsequently, we will update the sampling probabilities for each group based on Eq. (9). Following this step, we will sample users $B'$ from the batch $B$ in accordance with their respective sampling probabilities. Specifically, at step $t$, the ratio of the number of groups in $G_1$ and $G_0$ within $B'$ is as $1 - p^{(t)} : p^{(t)}$. Finally, we will utilize the query set loss ($L(f_{\theta_i}, D_i^q)$) of the sampled users $B'$ to update the fairness-aware meta-model parameters. The final optimization objective of the FAST can be represented as:

$$\min_\theta \sum_{u_i \in B'} L(f_{\theta_i}, D_i^q), \quad (10)$$
$$\text{where} \quad \theta_i = \theta - \alpha \nabla_{\theta_i} L(f_{\theta_i}, D_i^s).$$

## 5.5 Theoretical Guarantees

In this paper, we theoretically identify group proportion imbalance as a crucial factor leading to unfairness in meta-learning. To address this issue, we propose FAST, whose core idea is adaptively adjusting

the sampling probability of each group $p$. In this section, we present theoretical evidence illustrating that FAST can make $p$ converge to the optimal value while also revealing its potential to completely eliminate unfairness issues. In each optimization step, we can compute the fairness state of the meta-learning model optimized based on the current sampling probability $p^{(t)}$. Following the analysis of [25], we assume that the model has reached the optimal state of this step and denote the fairness result as $|\mathcal{L}_0^{\text{opt}}(p^{(t)}) - \mathcal{L}_1^{\text{opt}}(p^{(t)})|$, where $\mathcal{L}_0^{\text{opt}}(p^{(t)})$, $\mathcal{L}_1^{\text{opt}}(p^{(t)})$ represent the loss of groups $G_0$, $G_1$ obtained under the optimal model with the sampling probability $p^{(t)}$. Then, the fairness result function involving $p$ in FAST is expressed as $F(p) = |\mathcal{L}_0^{\text{opt}}(p) - \mathcal{L}_1^{\text{opt}}(p)|$. The optimal value $p^*$ of $F(p)$ can be represented as,

$$p^* = \arg\min_p F(p). \tag{11}$$

Subsequently, we attempt to answer the two questions theoretically:

- **Q1:** Can FAST make $p$ converge to the optimum value $p^*$?
- **Q2:** When $p$ reaches its optimum value $p^*$, is $F(p^*)$ equal to 0, implying that the meta model can achieve perfect fairness?

If the answers to both of the above questions are affirmative, then we can conclude that FAST has the potential to completely eliminate unfairness. For **Q1**, we will show that FAST can make the sampling probabilities of each group converge to the optimal values through Theorem 2 below. In other words, we aim to demonstrate that as long as the number of rounds $t$ is sufficiently large, the difference between sampling probability $p^{(t)}$ and the optimum value $p^*$ is bounded by a controllable parameter $\tau$. The detailed theoretical analysis is as follows, in Lemma 1 we will demonstrate that $F(p)$ is a quasi-convex function. Then, based on this, we prove an upper bound for $|p^{(t)} - p^*|$ in Theorem 2.

LEMMA 1. *Let* $F(p) = |\mathcal{L}_0^{\text{opt}}(p) - \mathcal{L}_1^{\text{opt}}(p)|$, *then we have,* $F(p)$ *is quasi-convex, i.e.,* $F(tp + (1-t)p') \leq \max\{F(p), F(p')\}$ *for all* $t \in [0, 1]$ *and* $p, p'$.

THEOREM 2. *We denote* $p^*$ *is the optimal value of* $F(p)$, *i.e.,* $p^* = \arg\min_p F(p)$, *then we have,*

$$|p^{(t)} - p^*| \leq \max\{|p^{(0)} - p^*| - t\tau, \tau\}. \tag{12}$$

REMARK. Specific proof details of Lemma 1 and Theorem 2 will be provided in the appendix A.2, A.3. Theorem 2 demonstrates that for a controllable parameter $\tau$, as long as the number of rounds $t$ is sufficiently large, $|p^{(0)} - p^*| - t\tau$ will always be less than $\tau$. This implies that $|p^{(t)} - p^*|$ is upper-bounded by a controllable parameter. Consequently, it indicates that the optimization of $p$ in FAST can converge to the optimum. □

For **Q2**, we will prove $F(p^*) = 0$ through Theorem 3,

THEOREM 3. *When* $p$ *reaches its optimal value* $p^*$, $F(p^*) = 0$.

The detailed proof is provided in Appendix A.4. Based on the responses to these questions, we can conclude that from a theoretical standpoint, FAST has the potential to entirely eliminate unfairness.

**Table 1: The statistics of the datasets.**

| Dataset | ML-1M | BookCrossing | ML-100K |
|---|---|---|---|
| #Users | 6,040 | 278,858 | 943 |
| #Items | 3,706 | 271,379 | 1,682 |
| #Ratings | 1,000,209 | 1,149,780 | 100,000 |
| Sparsity | 95.5316% | 99.9985% | 93.6953% |
| Sensitive Attribute | Gender | Age | Gender |
| Range of ratings | 1 - 5 | 1 - 10 | 1 - 5 |

## 6 EXPERIMENTS

In this section, we first introduce the dataset and experimental setup. Then, we conduct extensive experiments on real-world datasets to answer the following questions:

- **RQ1:** Does FAST learn fair and accurate results for cold-start problem?
- **RQ2:** How is the generalization ability of our proposed FAST on other cold-start meta-learning models?
- **RQ3:** How does the adaptive sampling impact FAST?
- **RQ4:** How do different hyper-parameter settings (i.e. $\gamma$, $\tau$) affect the fairness and utility performance?

The code is available for review[2] and will be open-sourced to support future research after this paper is accepted .

### 6.1 Datasets

In this paper, we conduct experiments using three publicly available recommender system datasets to validate the effectiveness of our FAST framework. We adopt the datasets ML-1M, BookCrossing, and ML-100K, following the precedent set by previous research [1, 17, 34]. Both the ML-1M and ML-100K datasets belong to the MovieLens Collection[3], containing movie rating data from users registered on the MovieLens website. Additionally, we utilize the BookCrossing dataset[4], a widely recognized benchmark for recommender systems that contains book rating data available on the web. These datasets encompass user, item, and their interaction information. Regarding the sensitive attribute, we select the gender attribute for the ML-100K and ML-1M datasets, while for the BookCrossing dataset, we adapt the age attribute as the sensitive attribute. For the sake of convenience, following [34], we categorize users into two groups based on whether they are over 40 years old. The dataset statistics are summarized in Table 1.

To evaluate user cold-start performance, we divided the users into training, validation, and testing datasets using a ratio of 7:1:2. In an effort to closely mimic real-world scenarios, following the setting in [17, 34], for each user $u_i$, we utilize a limited and variable number of interactions as their existing fine-tuned data $D_i^s$ while reserving the last 10 interactions as the query data $D_i^q$ for evaluation.

### 6.2 Experimental Setup and Baselines

*Evaluation.* In this paper, we evaluate the results of user modeling by predicting item ratings. This evaluation can be divided

---

[2]https://anonymous.4open.science/r/FAST-7516/
[3]https://grouplens.org/datasets/movielens/
[4]http://www2.informatik.uni-freiburg.de/ cziegler/BX/

**Table 2: The utility and fairness results on three datasets. The best fairness results on each backbone are highlighted in bold and underline represents the runner-up results. The lower, the better. All experiments are repeated five times on each dataset with mean and standard deviation results reported.**

| | ML-1M | | | | BookCrossing | | | | ML-100k | | | |
|---|---|---|---|---|---|---|---|---|---|---|---|---|
| | Utility | | Fairness | | Utility | | Fairness | | Utility | | Fairness | |
| | MAE ↓ | MSE ↓ | GF(MAE) ↓ | GF(MSE) ↓ | MAE ↓ | MSE ↓ | GF(MAE) ↓ | GF(MSE) ↓ | MAE ↓ | MSE ↓ | GF(MAE) ↓ | GF(MSE) ↓ |
| Wide&Deep | 0.937 ± 0.019 | 1.245 ± 0.026 | 0.120 ± 0.087 | 0.239 ± 0.169 | 1.558 ± 0.039 | 3.585 ± 0.356 | 0.083 ± 0.023 | 0.178 ± 0.188 | 1.071 ± 0.073 | 1.658 ± 0.203 | 0.164 ± 0.086 | 0.465 ± 0.274 |
| DropoutNet | 0.940 ± 0.028 | 1.347 ± 0.065 | 0.121 ± 0.065 | 0.271 ± 0.170 | 1.531 ± 0.034 | 3.523 ± 0.072 | 0.093 ± 0.051 | 0.184 ± 0.164 | 1.092 ± 0.063 | 1.694 ± 0.184 | 0.125 ± 0.061 | 0.358 ± 0.189 |
| NLBA | 1.060 ± 0.095 | 1.811 ± 0.309 | 0.051 ± 0.040 | 0.245 ± 0.188 | 1.659 ± 0.029 | 3.956 ± 0.071 | 0.103 ± 0.014 | 0.367 ± 0.029 | 0.999 ± 0.014 | 1.396 ± 0.036 | 0.109 ± 0.035 | 0.294 ± 0.100 |
| Melu | 0.796 ± 0.006 | 1.005 ± 0.018 | 0.057 ± 0.015 | 0.132 ± 0.045 | _1.464 ± 0.013_ | _3.458 ± 0.108_ | 0.056 ± 0.009 | 0.337 ± 0.114 | **0.850 ± 0.002** | **1.126 ± 0.003** | 0.089 ± 0.006 | 0.131 ± 0.014 |
| Melu + Reg | 0.790 ± 0.005 | 0.986 ± 0.022 | 0.060 ± 0.012 | 0.147 ± 0.025 | 1.522 ± 0.048 | 3.715 ± 0.177 | 0.088 ± 0.030 | 0.481 ± 0.118 | _0.875 ± 0.003_ | 1.193 ± 0.009 | 0.070 ± 0.009 | 0.150 ± 0.018 |
| Melu + IPW | 0.810 ± 0.009 | 1.035 ± 0.022 | 0.061 ± 0.012 | 0.156 ± 0.031 | 1.624 ± 0.127 | 4.051 ± 0.519 | 0.118 ± 0.114 | 0.543 ± 0.468 | 0.878 ± 0.008 | 1.171 ± 0.014 | 0.063 ± 0.008 | 0.084 ± 0.017 |
| Melu + CLOVER | **0.752 ± 0.004** | **0.906 ± 0.004** | _0.039 ± 0.003_ | 0.115 ± 0.008 | 1.652 ± 0.056 | 4.353 ± 0.233 | 0.110 ± 0.015 | 0.514 ± 0.074 | 0.875 ± 0.006 | **1.118 ± 0.014** | _0.055 ± 0.016_ | 0.066 ± 0.041 |
| Melu + FAST | _0.763 ± 0.003_ | _0.933 ± 0.007_ | **0.033 ± 0.004** | **0.108 ± 0.009** | **1.473 ± 0.009** | **3.369 ± 0.018** | **0.017 ± 0.009** | **0.168 ± 0.053** | 0.898 ± 0.004 | 1.217 ± 0.015 | **0.044 ± 0.006** | **0.056 ± 0.022** |
| MetaCS | 0.774 ± 0.007 | 0.959 ± 0.012 | 0.050 ± 0.005 | 0.148 ± 0.014 | 1.445 ± 0.013 | 3.420 ± 0.033 | 0.088 ± 0.011 | 0.436 ± 0.059 | _0.905 ± 0.005_ | 1.318 ± 0.004 | 0.092 ± 0.003 | 0.187 ± 0.006 |
| MetaCS + Reg | 0.797 ± 0.003 | 0.995 ± 0.005 | _0.026 ± 0.009_ | _0.080 ± 0.006_ | **1.327 ± 0.007** | **2.894 ± 0.031** | 0.065 ± 0.015 | 0.298 ± 0.045 | 0.912 ± 0.003 | 1.314 ± 0.007 | 0.078 ± 0.002 | 0.174 ± 0.010 |
| MetaCS + IPW | **0.760 ± 0.002** | **0.916 ± 0.006** | 0.046 ± 0.006 | 0.124 ± 0.007 | 1.399 ± 0.011 | 3.216 ± 0.057 | 0.063 ± 0.013 | 0.190 ± 0.033 | 0.920 ± 0.008 | 1.329 ± 0.010 | 0.065 ± 0.009 | 0.208 ± 0.011 |
| MetaCS + CLOVER | 0.804 ± 0.036 | 1.027 ± 0.095 | 0.049 ± 0.011 | 0.133 ± 0.017 | 1.973 ± 0.332 | 5.851 ± 1.239 | _0.029 ± 0.019_ | _0.168 ± 0.133_ | 0.918 ± 0.022 | _1.304 ± 0.081_ | **0.043 ± 0.020** | **0.083 ± 0.021** |
| MetaCS + FAST | _0.770 ± 0.003_ | _0.938 ± 0.007_ | **0.026 ± 0.006** | **0.072 ± 0.007** | _1.385 ± 0.014_ | _3.204 ± 0.055_ | **0.025 ± 0.011** | **0.117 ± 0.083** | 0.879 ± 0.003 | 1.184 ± 0.011 | 0.054 ± 0.002 | _0.118 ± 0.010_ |

into two main aspects: utility evaluation and fairness evaluation. For utility evaluation, we employ classical metrics MAE, MSE. For fairness evaluation, we adopt the User-oriented Group Fairness Definition introduced in Section 3.2. For the $\mathcal{M}$ in Eq. (2), we adopt MAE, MSE. In our final experiments, we will use GF(MAE) and GF(MSE) to represent these fairness metrics. Lower GF indicates better fairness performance.

*Implementation detail.* We incorporate two meta-learning backbones, MELU and MetaCS, to illustrate the generalization of the proposed framework. For each backbone, we evaluate the fairness on the query set and adjust the sample probability $p$ based on the evaluated fairness. Following [17], We adopt two-layer network as our user preference estimator, and each layer has 64 nodes. We set the dimension of all embedding vectors to 32 and adopt the mean square error (mse) as our loss function. Adam algorithm is used to optimize the model. We set the number of training epochs to 50 and train 32 user tasks in one batch. The parameter configuration of FAST is shown in Appendix A.5 (Table 3). We implement all models with PyTorch and conduct all experiments on four 2.0GHz Intel Xeon E5-2620 CPUs and a Tesla K20m GPU.

*Baseline approaches.* FAST is a fairness-aware framework that can be applied to various meta-learning paradigms. In this paper, following [34], we utilize FAST in two well-established meta-learning backbones:

- MELU [17]: MELU is a personalized user preference estimation model based on meta-learning, designed for rapid adaptation to new users.
- MetaCS [2]: MetaCS shares a comparable conceptual foundation with MELU in the construction of its recommender model. Furthermore, it leverages a highly adaptable meta-update strategy for the acquisition of model parameters.

To validate the effectiveness of FAST, we integrate several classic fairness-aware methods into the realm of meta-learning:

- Reg [39]: Reg is a regularization-based approach. Here, we incorporates Eq. (2) into the outer-loop loss function.
- IPW [22]: IPW utilizes standard inverse propensity weights to reweight samples, mitigating fairness issues.
- CLOVER [34]: CLOVER serves as a state-of-the-art fairness baseline for meta-learning based on adversarial learning.

Furthermore, we conduct a comparative analysis involving various classic cold-start baselines to demonstrate FAST's effectiveness in cold-start scenarios:

- Wide & Deep [7]: Wide & Deep predicts user preferences for items using a deep neural network, employing the same neural network architecture as MELU.
- DropoutNet [29]: DropoutNet combines the dropout technique with a deep neural network to effectively learn input features for addressing cold-start problems.
- NLBA [28] : NLBA trains a cold-start neural network recommender, which maintains constant weights (both output and hidden) across users, adapting only the biases of output and hidden units on a per-user basis.

## 6.3 Experimental Results

*Overall Results (RQ1, RQ2).* In this section, we delve into whether FAST can achieve fair and accurate results. Furthermore, FAST represents a general framework that can be applied to different meta-learning methods (e.g., Melu and MetaCS). We also explore the generalization of FAST. The utility and fairness results on three datasets are presented in Table 2. From these results, we can draw the following findings:

- From the perspective of the cold-start problem, we observe that meta-learning methods exhibit significant performance improvements compared to traditional cold-start approaches. This underscores the superiority of the meta-learning paradigm. Additionally, we uncover instances of unfairness in all cold-start methods, underscoring the necessity of investigating fairness in cold-start scenarios.
- From the perspective of utility, we are surprised to discover that many fairness-aware methods, such as CLOVER, can even enhance the performance of the original methods, aligning with their papers' claims [34], which argues that adversarial learning may aid in achieving better convergence towards optimality. Furthermore, our utility results consistently yield comparable outcomes, and in some cases, our approach outperforms traditional meta-learning methods. We believe this is because our process may filter out redundant information during the sampling phase, thereby facilitating improved meta-learning convergence.

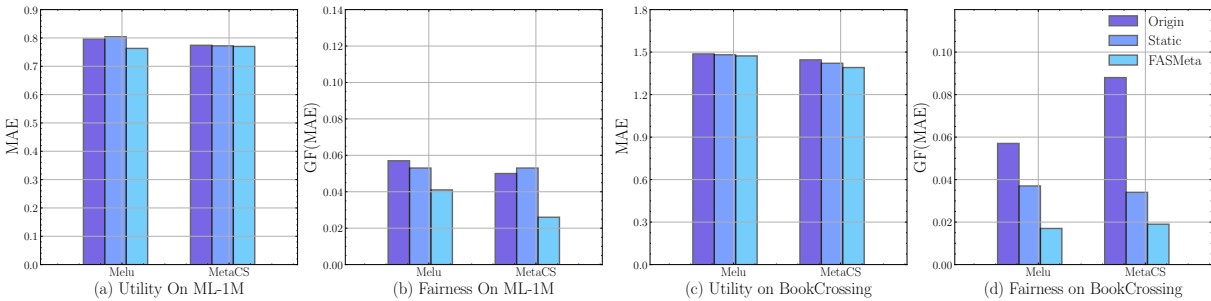

Figure 3: The utility and fairness result on different sample strategies (the lower, the better).

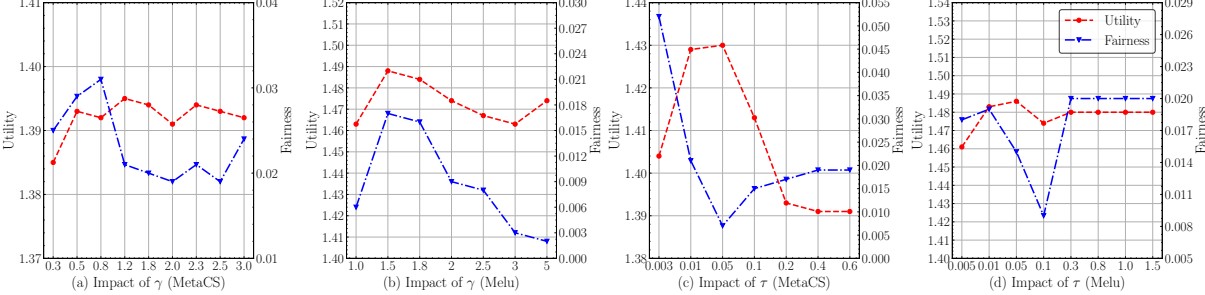

Figure 4: The impact of $\gamma$, $\tau$ for different meta-learning backbones on the BookCrossing dataset (the lower, the better).

- From the perspective of fairness, we find that in two meta-learning paradigms, the majority of fairness-aware methods enhance fairness, with FAST showing the most significant improvement. This highlights the effectiveness of FAST in addressing fairness concerns across different paradigms.

In summary, our FAST demonstrates satisfactory fairness results and comparable utility across various meta-learning paradigms, showcasing the effectiveness and generality of FAST.

*Impact of Adaptive Sampling (RQ3).* Our paper theoretically identifies group proportion imbalance as a pivotal factor contributing to unfairness in meta-learned systems. To efficiently address this factor, we propose an adaptive sampling strategy. In this section, we validate the superiority of this adaptive strategy on ML-1M and BookCrossing datasets. Specifically, we compare it to the simplest static sampling approach (referred to as **Static**), where the probability of sampling different groups is equal in each step. The results are depicted in Figure 3. Here, **Origin** refers to the meta-learning strategy without any fairness considerations. From the results, we can observe that both the Static and FAST strategies achieve improved fairness performance and comparable utility performance when compared to the original meta-learning paradigm. This suggests that sampling strategy can mitigate the unfairness stemming from group proportion imbalance to a certain extent. Furthermore, we find that FAST outperforms Static in terms of fairness performance, demonstrating the superiority of the adaptive sampling strategy.

*Effect of Hyper-parameters (RQ4).* In FAST, there are two hyper-parameters that control the effectiveness of FAST. Specifically, $\gamma$ controls the rate of change in sampling probabilities, and $\tau$ sets an upper threshold for the change in sampling probabilities. In this section, we examine the impact of adjusting these two hyperparameters on FAST using the BookCrossing dataset. The specific results are shown in Figure 4. Figure 4 (a) and Figure 4 (b) illustrate

the effects of $\gamma$ on different meta-learning backbones. Overall, they exhibit similar trends. When $\gamma$ initially increases, both fairness and utility deteriorate. However, after reaching a certain threshold (0.8 for MetaCS, 1.5 for Melu), fairness improves, and the change in accuracy becomes more gradual. This phenomenon indicates that increasing $\gamma$ has a beneficial impact on the fairness of FAST and can aid in achieving a trade-off between fairness and utility. Figures 4 (c) and Figure 4 (d) depict the effects of $\tau$ on different backbones. We observe that initially increasing $\tau$ enhances fairness. This demonstrates that setting $\tau$ can help FAST achieve better fairness results. However, when $\tau$ reaches a certain threshold (0.05 for MetaCS, 0.1 for Melu), fairness begins to degrade. As $\tau$ increases further, FAST's performance remains relatively stable. We believe this is because when $\tau$ becomes too large, the changes in model sampling probabilities consistently fall below this threshold, rendering $\tau$ ineffective and no longer beneficial for the model.

## 7 CONCLUSION

In this paper, we presented a focused study on fairness in meta-learning for user modeling. Specifically, through a rigorous theoretical analysis that integrated the meta-learning paradigm with the classical fairness metric, we identified group proportion imbalance as a pivotal factor that contributed to unfairness. In light of this insight, we proposed a novel fairness-aware adaptive sampling framework for meta-learning, known as FAST, which could adaptively adjust the group-level sampling distribution during the interleaved training process. Moreover, we provided theoretical proof demonstrating FAST's potential to completely eliminate unfairness. Finally, we conducted experiments on three datasets and the results demonstrated that FAST could effectively enhances fairness while maintaining high accuracy. In the future, we intend to expand our exploration of fairness into other paradigms, such as contrastive learning, to further enhance our understanding and development of fair machine learning techniques.

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

# A APPENDIX

## A.1 Theorem 1

THEOREM 1. *Given the user group $U$, comprising $M$ users, which can be divided into two subgroups, $G_0$ and $G_1$, based on binary sensitive attributes. Let the number of users in each group be represented as $m_0$ and $m_1$, respectively. Without loss of generality, let us assume $m_1 > m_0$. We further assume the actual preference of user $u_i$ is $x_i$,*

*and the actual preferences of users in group $G_0$ and group $G_0$ are drawn from two normal distributions, i.e., $\mathcal{N}(-\eta, \sigma^2)$ and $\mathcal{N}(\eta, \sigma^2)$, respectively. Then we have: with probability at least $1 - \frac{A}{\frac{m_1 - m_0}{2M}}$, the loss gap $\mathcal{L}_1 - \mathcal{L}_0$ satisfies:*

$$\mathcal{L}_1 - \mathcal{L}_0 > \frac{m_1 - m_0}{2M}\eta^2 C, \tag{13}$$

*where $C$ is a constant related to the learning rate $\alpha$, and $A = C^2[2(1 - \frac{1}{M})^2\sigma^4(\frac{1}{m_0} + \frac{1}{m_1}) + 16(1 - \frac{1}{M})\eta^2(\frac{m_0^3 + m_1^3}{m_0 m_1 M^2})]$.*

PROOF. Given that the actual preference of user $u_i$ is $x_i$, we can express $L(f_{\theta_i}, D_i^q)$ as $(\theta_i - x_i)^2$. For the sake of convenience, we use $L_i$ to represent this expression. Then the primary objective of meta-learning can be represented as follows:

$$\min_\theta \sum_{u_i \in B} L_i \tag{14}$$
$$\text{where} \quad \theta_i = \theta - \alpha\nabla_{\theta_i} L_i$$

Then, express the total loss $L$ as the sum of losses from group 0 and group 1, and we have:

$$L = \sum_{u_i \in B} L_i$$
$$= \sum_{u_i \in G_0} (\theta_i - x_i)^2 + \sum_{u_k \in G_1} (\theta_k - x_k)^2. \tag{15}$$

Taking the gradient of $L_i$, we get

$$\nabla_{\theta_i} L_i = \nabla_{\theta_i}(\theta_i - x_i)^2$$
$$= 2(\theta_i - x_i). \tag{16}$$

Substituting the gradient part in Eq. (14) with Eq. (16) and combining Eq. (15), we get:

$$L = \sum_{i \in G_0} [(1 - 2\alpha)(\theta - x_i)]^2 + \sum_{i \in G_1} [(1 - 2\alpha)(\theta - x_i)]^2. \tag{17}$$

Then we can solve the problem in Eq. (14) by solving $\nabla_\theta L = 0$. Therefore, we have

$$\theta^* = \frac{m_0}{m_0 + m_1}\bar{x}_0 + \frac{m_1}{m_0 + m_1}\bar{x}_1, \tag{18}$$

where $\theta^*$ is the optimum solution of Eq. (14) and $\bar{x}_0$ and $\bar{x}_1$ are the centroid of samples in group 0 and group 1 respectively.

Now we define $\mathcal{L}_0 = \frac{1}{m_0}\sum_{u_i \in G_0}(\theta_i - x_i)^2$ as the average loss of users in group 0. Putting $\theta^*$ into $\mathcal{L}_0$ and $\mathcal{L}_1$, we obtain

$$\mathcal{L}_0 = (1 - 2\alpha)^2 \frac{1}{m_0}\sum_{u_i \in G_0}(\theta^* - x_i)^2 \tag{19}$$

For the sake of simplicity, we denote $C$ as $(1 - 2\alpha)^2$ and $\mathcal{L}_0'$ as $\frac{1}{m_0}\sum_{u_i \in G_0}(\theta^* - x_i)^2$ respectively. In similar way, we can define $\mathcal{L}_1$ and $\mathcal{L}_1'$.

If we view $x_i$ as a random variable that drawn from some normal distributions, then the average loss $\mathcal{L}_0$ is also a random variable, as well as $\mathcal{L}_1$.

We denote $\mathcal{L}_{\text{gap}}$ as the average loss gap between group 0 and group 1, i.e.,

$$\mathcal{L}_{\text{gap}} = \mathcal{L}_0 - \mathcal{L}_1 = C * (\mathcal{L}_0' - \mathcal{L}_1'). \tag{20}$$

To prove the theorem, we first calculate the expectation and variance of $\mathcal{L}_0'$. And the expectation and variance of $\mathcal{L}_1'$ can be

calculated similarly. We let $Y_i = \theta^* - x_i$ and put Eq. (18) into its expression, then we have:

$$Y_i = \theta^* - x_i$$
$$= \frac{m_0}{m_0 + m_1}\frac{1}{m_0}\sum_{u_j \in G_0} x_j + \frac{m_1}{m_0 + m_1}\frac{1}{m_1}\sum_{u_j \in G_1} x_j - x_i$$
$$= \frac{1}{m_0 + m_1}\sum_{\substack{u_j \in G_0 \\ j \neq i}} x_j + \frac{1}{m_0 + m_1}\sum_{j \in G_1} x_j - (1 - \frac{1}{m_0 + m_1})x_i. \tag{21}$$

Since all $x_j$ are normal and independent, $Y_i$ is also normal. According to the properties of expectation and variance, we have

$$\mathbb{E}[Y_i] = -\frac{2m_1}{m_0 + m_1}\eta,$$

and

$$\text{Var}[Y_i] = [1 - \frac{1}{m_0 + m_1}]\sigma_0^2.$$

Hence, we get

$$Y_i \sim \mathcal{N}(-\frac{2m_1}{m_1 + m_0}\eta, \ [1 - \frac{1}{m_0 + m_1}]\sigma_0^2). \tag{22}$$

Assume a random variable $z$ is drawn from standard normal distribution, then $Y_i = \sqrt{1 - \frac{1}{m_0 + m_1}}\sigma_0 z + \frac{2m_1}{m_0 + m_1}\eta$ and

$$Y_i^2 = [1 - \frac{1}{m_0 + m_1}]\sigma_0^2 z^2 + 4[\frac{m_1}{m_0 + m_1}]^2\eta^2 +$$
$$4\sqrt{1 - \frac{1}{m_0 + m_1}}\frac{m_1}{m_0 + m_1}\eta \cdot \sigma_0 z. \tag{23}$$

Take the expectation on both sides of Eq. (23), we get the expectation of the $\mathcal{L}_0'$:

$$\mathbb{E}[\mathcal{L}_0'] = \mathbb{E}[Y_i^2]$$
$$= \frac{m_1^2}{(m_1 + m_0)^2}\eta^2 + [1 - \frac{1}{m_0 + m_1}]\sigma_0^2. \tag{24}$$

Similarly, take the variance on both sides of Eq. (23), we get the variance of $\mathcal{L}_0'$:

$$\text{Var}[\mathcal{L}_0'] = \frac{1}{m_0}\text{Var}[Y_i^2]$$
$$= \frac{2}{m_0}(1 - \frac{1}{m_0 + m_1})^2\sigma_0^4 + \frac{16}{m_0}(1 - \frac{1}{m_0 + m_1})(\frac{m_1}{m_0 + m_1})^2\eta^2. \tag{25}$$

The expectation and variance of $\mathcal{L}_1'$ can be calculated in the same way:

$$\mathbb{E}[\mathcal{L}_1'] = \frac{m_0^2}{(m_1 + m_0)^2}\eta^2 + [1 - \frac{1}{m_0 + m_1}]\sigma_1^2, \tag{26}$$

and

$$\text{Var}[\mathcal{L}_1'] = \frac{1}{m_1}\text{Var}[Y_i^2]$$
$$= \frac{2}{m_1}(1 - \frac{1}{m_0 + m_1})^2\sigma_1^4 + \frac{16}{m_1}(1 - \frac{1}{m_0 + m_1})(\frac{m_0}{m_0 + m_1})^2\eta^2. \tag{27}$$

As a result, we have

$$\mathbb{E}[\mathcal{L}_{\text{gap}}] = \mathbb{E}[C(\mathcal{L}_0' - \mathcal{L}_1')]$$
$$= C * [\mathbb{E}[\mathcal{L}_0'] - \mathbb{E}[\mathcal{L}_1']] \tag{28}$$

and

$$\text{Var}[\mathcal{L}_{\text{gap}}] = C^2 [2(1 - \frac{1}{m_0 + m_1})^2 \sigma^4 (\frac{1}{m_0} + \frac{1}{m_1})$$
$$+ 16(1 - \frac{1}{m_0 + m_1}) \eta^2 (\frac{m_0^3 + m_1^3}{m_0 m_1 (m_0 + m_1)^2})]. \quad (29)$$

From Chebyshev's inequality, for any $\epsilon > 0$, we have:

$$\Pr[|\mathcal{L}_{\text{gap}} - \mathbb{E}[\mathcal{L}_{\text{gap}}]| < \epsilon] > 1 - \frac{\text{Var}[\mathcal{L}_{\text{gap}}]}{\epsilon^2}, \quad (30)$$

i.e.,

$$\Pr[\mathbb{E}[\mathcal{L}_{\text{gap}}] - \epsilon < \mathcal{L}_{\text{gap}} < \mathbb{E}[\mathcal{L}_{\text{gap}}] + \epsilon] > 1 - \frac{\text{Var}[\mathcal{L}_{\text{gap}}]}{\epsilon^2}. \quad (31)$$

Since $\Pr[\mathbb{E}[\mathcal{L}_{\text{gap}}] - \epsilon < \mathcal{L}_{\text{gap}}] > \Pr[\mathbb{E}[\mathcal{L}_{\text{gap}}] - \epsilon < \mathcal{L}_{\text{gap}} < \mathbb{E}[\mathcal{L}_{\text{gap}}] + \epsilon]$, we have

$$\Pr[\mathbb{E}[\mathcal{L}_{\text{gap}}] - \epsilon < \mathcal{L}_{\text{gap}}] > 1 - \frac{\text{Var}[\mathcal{L}_{\text{gap}}]}{\epsilon^2}. \quad (32)$$

Let $\epsilon = \frac{1}{2}\mathbb{E}[\mathcal{L}_{\text{gap}}]$, we prove the theorem. □

## A.2 Lemma 1

LEMMA 1. *$F(p)$ is quasi-convex, i.e., $F(tp+(1-t)p') \leq \max\{F(p), F(p')\}$ for all $t \in [0, 1]$ and $p, p'$.*

PROOF. Before prove the theorem, we first express our objective as follows:

$$\min_p |\mathcal{L}_0^{\text{opt}}(p) - \mathcal{L}_1^{\text{opt}}(p)| \quad (33)$$

where $\mathcal{L}_0^{\text{opt}}(p)$ and $\mathcal{L}_1^{\text{opt}}(p)$ are the average losses on group 0 and 1, respectively, of the optimal model trained with sample probability $p$. Hence, they satisfy the following inequality:

$$p\mathcal{L}_0^{\text{opt}}(p) + (1-p)\mathcal{L}_1^{\text{opt}}(p) \leq p\mathcal{L}_0 + (1-p)\mathcal{L}_1, \quad \forall\theta \quad (34)$$

As known in [4], a continual function $f : \mathbb{R} \rightarrow \mathbb{R}$ is quasi-convex if and only if at least of one of the following conditions holds: 1) non-decreasing, 2) non-increasing, 3) non-increasing and then non-decreasing. As a result, we can prove the function $F(p)$ is quasi-convex by showing that it is first non-increasing and then non-decreasing.

To show this, we can first prove that $\mathcal{L}_0^{\text{opt}}(p) - \mathcal{L}_1^{\text{opt}}(p)$ is non-increasing. After that, the conclusion that $F(p)$ is non-increasing and then non-decreasing can be easily proved.

Considering $p_a > p_b$, if we can prove that $\mathcal{L}_0^{\text{opt}}(p_a) \leq \mathcal{L}_0^{\text{opt}}(p_b)$ and $\mathcal{L}_1^{\text{opt}}(p_a) \geq \mathcal{L}_1^{\text{opt}}(p_b)$, then it is obvious that $\mathcal{L}_1^{\text{opt}}(p) - \mathcal{L}_0^{\text{opt}}(p)$ is non-increasing. We prove this by showing that it is only the cases that do not contradict known conditions. From Eq. (34) we can derive following conditions:

$$p_a\mathcal{L}_0^{\text{opt}}(p_a) + (1-p_a)\mathcal{L}_1^{\text{opt}}(p_a) \leq p_a\mathcal{L}_0 + (1-p_a)\mathcal{L}_1, \forall\theta, \quad (35a)$$
$$p_b\mathcal{L}_0^{\text{opt}}(p_b) + (1-p_b)\mathcal{L}_1^{\text{opt}}(p_b) \leq p_b\mathcal{L}_0 + (1-p_b)\mathcal{L}_1, \forall\theta. \quad (35b)$$

We prove $\mathcal{L}_0^{\text{opt}}(p_a) \leq \mathcal{L}_0^{\text{opt}}(p_b)$ and $\mathcal{L}_1^{\text{opt}}(p_a) \geq \mathcal{L}_0^{\text{opt}}(p_b)$ by showing that other cases contradict Eq. (35):

- **Case 1** $\mathcal{L}_0^{\text{opt}}(p_a) > \mathcal{L}_0^{\text{opt}}(p_b)$ and $\mathcal{L}_1^{\text{opt}}(p_a) \geq \mathcal{L}_1^{\text{opt}}(p_b)$:
  By multiplying these two inequalities with their corresponding coefficients $p_a$ and $1 - p_a$, we get:

$$p_a\mathcal{L}_0^{\text{opt}}(p_a) + (1-p_a)\mathcal{L}_1^{\text{opt}}(p_a) > p_a\mathcal{L}_0^{\text{opt}}(p_b) + (1-p_a)\mathcal{L}_1^{\text{opt}}(p_b),$$

  which is contradicted with Eq. (35a).
- **Case 2** $\mathcal{L}_0^{\text{opt}}(p_a) \leq \mathcal{L}_0^{\text{opt}}(p_b)$ and $\mathcal{L}_1^{\text{opt}}(p_a) < \mathcal{L}_1^{\text{opt}}(p_b)$:
  By multiplying these two inequalities with their corresponding coefficients $p_b$ and $1 - p_b$, we get:

$$p_b\mathcal{L}_0^{\text{opt}}(p_b) + (1-p_b)\mathcal{L}_1^{\text{opt}}(p_b) > p_b\mathcal{L}_0^{\text{opt}}(p_a) + (1-p_b)\mathcal{L}_1^{\text{opt}}(p_a),$$

  which is contradicted with Eq. (35b).
- **Case 3** $\mathcal{L}_0^{\text{opt}}(p_a) > \mathcal{L}_0^{\text{opt}}(p_b)$ and $\mathcal{L}_1^{\text{opt}}(p_a) < \mathcal{L}_1^{\text{opt}}(p_b)$:
  Add Eq. (35a) with $\mathcal{L}_0 = \mathcal{L}_0^{\text{opt}}(p_b)$ and $\mathcal{L}_1 = \mathcal{L}_1^{\text{opt}}(p_b)$, and Eq. (35b) with $\mathcal{L}_0 = \mathcal{L}_0^{\text{opt}}(p_a)$ and $\mathcal{L}_1 = \mathcal{L}_1^{\text{opt}}(p_a)$, we have:

$$p_a\mathcal{L}_0^{\text{opt}}(p_a) + (1-p_a)\mathcal{L}_1^{\text{opt}}(p_a) + p_b\mathcal{L}_0^{\text{opt}}(p_b) + (1-p_b)\mathcal{L}_1^{\text{opt}}(p_b)$$
$$\leq p_a\mathcal{L}_0^{\text{opt}}(p_b) + (1-p_a)\mathcal{L}_1^{\text{opt}}(p_b) + p_b\mathcal{L}_0^{\text{opt}}(p_a) + (1-p_b)\mathcal{L}_1^{\text{opt}}(p_a). \quad (36)$$

  Rearrange the items, we get

$$(p_a-p_b)[\mathcal{L}_0^{\text{opt}}(p_a)-\mathcal{L}_0^{\text{opt}}(p_b)] \leq (p_a-p_b)[\mathcal{L}_1^{\text{opt}}(p_a)-\mathcal{L}_1^{\text{opt}}(p_b)]. \quad (37)$$

  Since we assume $p_a > p_b$, we can divide the both side of equation with $(p_a - p_b)$ and get

$$\mathcal{L}_0^{\text{opt}}(p_a) - \mathcal{L}_0^{\text{opt}}(p_b) \leq \mathcal{L}_1^{\text{opt}}(p_a) - \mathcal{L}_1^{\text{opt}}(p_b), \quad (38)$$

  which contradicts the conditions of $\mathcal{L}_1^{\text{opt}}(p_b) > \mathcal{L}_1^{\text{opt}}(p_a)$ and $\mathcal{L}_0^{\text{opt}}(p_a) > \mathcal{L}_0^{\text{opt}}(p_b)$.

By showing that all other cases contradict Eq. (35), we prove $\mathcal{L}_0^{\text{opt}}(p_a) \leq \mathcal{L}_0^{\text{opt}}(p_b)$ and $\mathcal{L}_1^{\text{opt}}(p_a) \geq \mathcal{L}_1^{\text{opt}}(p_b)$ holds, which implies the quasi-convexity of $F(p)$ as mentioned earlier. Hence we complete the proof.

□

## A.3 Theorem 2

THEOREM 2. *We denote $p^*$ is the optimal value of $F(p)$, i.e., $p^* = \arg_p \min F(p)$, then we have,*

$$|p^{(t)} - p^*| \leq \max\{|p^{(0)} - p^*| - t\tau, \tau\} \quad (39)$$

PROOF. As shown before, $\mathcal{L}_0^{\text{opt}}(p) - \mathcal{L}_1^{\text{opt}}(p)$ is non-increasing and can reach 0. Therefore, we have $\mathcal{L}_0^{\text{opt}}(p) - \mathcal{L}_1^{\text{opt}}(p) \geq 0 \ \forall p \in [0, p^*]$ and $\mathcal{L}_0^{\text{opt}}(p) - \mathcal{L}_1^{\text{opt}}(p) \leq 0 \ \forall p \in [p^*, 1]$, where $p^*$ is the zero point that satisfies $\mathcal{L}_0^{\text{opt}}(p^*) - \mathcal{L}_1^{\text{opt}}(p^*) = 0$

Then, we complete our proof by considering two cases:

- **Case 1** For all $i \in \{0, 1, 2, \cdots k\}$, $p^{(i)} > p^*$ holds or for all $i \in \{0, 1, 2, \cdots k\}$, $p^{(i)} < p^*$ holds. In this case, $\forall i \in \{0, 1, 2, \cdots k\}$, the sign$(\mathcal{L}_0^{\text{opt}}(p^{(i)}) - \mathcal{L}_1^{\text{opt}}(p^{(i)}))$ is the same. Then according to the update rule, we have

$$|p^{(k)} - p^*| = |p^{(0)} - p^*| - \sum_{i=0}^{k} \min\{\gamma|\mathcal{L}_0^{\text{opt}}(p^{(i)}) - \mathcal{L}_1^{\text{opt}}(p^{(i)})|, \tau\} \quad (40)$$

If there exist some $i$ such that $|\mathcal{L}_1^{\text{opt}}(p^{(i)}) - \mathcal{L}_0^{\text{opt}}0(p^{(i)})| = 0$, then we have got an optimal solution $p^{(i)}$. Otherwise, we can find a $\tau > 0$ such that $\gamma|\mathcal{L}_1^{\text{opt}}(p^{(i)}) - \mathcal{L}_0^{\text{opt}}(p^{(i)})| > \tau$. In this case, we can rewrite 40 as:

$$|p^{(k)} - p^*| = |p^{(0)} - p^*| - k\tau \tag{41}$$

- **Case 2** $\exists j \in \{0, 1, 2, \cdots k - 1\}$, let $p^{(j)} < p^* < p^{(j+1)}$ or $p^{(j)} > p^* > p^{(j+1)}$.
  In this case, we prove $|p^{(k)} - p^*| \leq \tau$ by showing that $|p^{(k)} - p^*| > \tau$ contradicts our condition. We assume $p^{(k)} > p^*$ (The other case of $p^{(k)} < p^*$ can be proved in the same way).
  If $p^{(k-1)} > p^{(k)}$, then we have $|p^{(k-1)} - p^*| = p^{(k-1)} - p^* > p^{(k)} - p^* > \tau$.
  If $p^* < p^{(k-1)} < p^{(k)}$, then we have $\mathcal{L}_0^{\text{opt}}(p^{(k-1)}) - \mathcal{L}_1^{\text{opt}}(p^{(k-1)}) < 0$. According to the update rule, we can derive that $p^{(k-1)} > p^{(k)}$, which is contradict to our condition. If $p^{(k-1)} < p^* < p^{(k)}$ then we have $|p^{(k)} - p^*| < |p^{(k)} - p^{(k-1)}| \leq \tau$, which is contradict to our assumption.
  Considering the above three cases, we can derive that if $|p^{(k)} - p^*| > \tau$ holds, then $|p^{(k-1)} - p^*| > \tau$ also holds. As a result, $|p^{(i)} - p^*| > \tau$ must holds for all $i \in \{0, 1, 2, \cdots, k\}$. However, this contradicts our condition since from the condition there $\exists j \in \{0, 1, 2, \cdots, k - 1\}$ letting $|p^{(j)} - p^*| < |p^{(j)} - p^{(j+1)}| \leq \tau$.

To summarize, in this case, $|p^{(k)} - p^*| \leq \tau$ holds.

Combining Case 1 and Case 2, we prove the theorem. □

## A.4 Theorem 3

THEOREM 3. *When $p$ reaches its optimal value $p^*$, $F(p^*) = 0$.*

PROOF. We assume that $\mathcal{L}_0^{\text{opt}}(p) - \mathcal{L}_1^{\text{opt}}(p)$ is not always negative or positive. This assumption is reasonable since it is almost impossible that the average loss of a group is always bigger than that of the other group no matter how the training users are sampled. Based on this assumption, according to the intermediate value theorem, there is a point $p^*$ such that $\mathcal{L}_0^{\text{opt}}(p^*) - \mathcal{L}_1^{\text{opt}}(p^*) = 0$. As a result, we have $F(p^*) = 0$ where $p^*$ is the optimal value of $F(p)$.

□

## A.5 Parameter Configuration

**Table 3: The parameter configuration of FAST.**

|  | ML-1M | | BookCrossing | | ML-100K | |
| --- | --- | --- | --- | --- | --- | --- |
|  | $\gamma$ | $\tau$ | $\gamma$ | $\tau$ | $\gamma$ | $\tau$ |
| Melu | 0.3 | 0.8 | 0.5 | 0.1 | 2 | 0.2 |
| MetaCS | 2 | 0.2 | 0.3 | 0.8 | 2 | 0.2 |