# OpenReview forum: "Enhancing Fairness in Meta-learned User Modeling via Adaptive Sampling"
_ACM.org/TheWebConf/2024/Conference — TheWebConf24_

### Official Review · Reviewer_U3wk · 2023-11-17

**Novelty:** 6
**Technical Quality:** 6

**Review:**

The study focus on the fairness in meta learning for user modeling. The authors identify the proportion imbalance is an important factor to the unfairness in meta learning, and propose fairness-aware adaptive sampling framework for meta-learning, name FAST, and provide the theoretical proof that FAST can eliminate the unfairness. The experimental results also demonstrate the effectiveness of proposed methods.


Advantages:
1.	The fairness in meta learning and its critical factor are important research questions.
2.	The study finds that group proportion imbalance during the outer loop is a critical determinant of fairness in meta learning.
3.	The authors also propose a novel framework to improving the fairness in meta learning, and offer the theoretical evidence.
4.	In the experiments, the authors compare the proposed method with many baseline methods with three datasets.

Improvement:

I suggest the authors verify their FAST method with more datasets like Last.fm, “Algorithmic fairness datasets: the story so far” introduce some datasets about fairness study.

**Questions:**

I suggest the authors verify their FAST method with more datasets like Last.fm, “Algorithmic fairness datasets: the story so far” introduce some datasets about fairness study.

**Reviewer Confidence:**

3: The reviewer is confident but not certain that the evaluation is correct

**Scope:**

4: The work is relevant to the Web and to the track, and is of broad interest to the community

---

### Official Review · Reviewer_PYWE · 2023-11-25

**Novelty:** 4
**Technical Quality:** 4

**Review:**

This paper addresses the problem of group fairness using meta-learning and provides a detailed theoretical analysis of the relationship between group fairness and the sizes of different groups. The experiments conducted demonstrate that the proposed algorithm outperforms the majority of baseline methods.

**Questions:**

Overall, it is an interesting direction. However, I have a few questions regarding this paper:

(1) Firstly, there seems to be a similarity between the outloop and inloop of meta-learning and the feedback loop in fairness, where using predicted results for further training may cause the model to favor a larger group. What are the differences between meta-learning and these feedback loops? It would be helpful to discuss this further.

(2) The assumptions of homoscedasticity in Theorem 1 appear to be strong. Could you please discuss this?

(3) One question is, if the cause of unfairness is the difference in group sizes, why not use weighted and resampling techniques to align their distributions? Could you discuss this approach?

(4) In the experiments, it would be beneficial to include how the model performs when the degree of distributional differences between groups varies and compare it with other methods that alter the distributions.

**Ethics Review Description:**

-

**Reviewer Confidence:**

3: The reviewer is confident but not certain that the evaluation is correct

**Scope:**

3: The work is somewhat relevant to the Web and to the track, and is of narrow interest to a sub-community

---

### Official Review · Reviewer_snFK · 2023-11-25

**Novelty:** 4
**Technical Quality:** 5

**Review:**

To solve the group proportion imbalance problem in terms of meta-learning based user modeling, the authors propose a framework named FAST which adaptively adjusts the sampling distribution for different user groups during the meta-training process. Theoretical analysis as well as empirical results are shown to demonstrate the effectiveness of the proposed method.

+ The manuscript is generally well-written with detailed description for the motivations as well as the illustrations for the proposed framework, which help the readers better understand the contributions.

+ Adjusting the sampling probabilities for different user groups to enhance the fairness is intuitive and easy to adopt.

+ Theoretical analysis is provided to demonstrate the effectiveness as well as the convergence of the proposed methods.

- Although the improvement over existing method is reasonable in terms of the fairness levels, the utility performance is often inferior to existing works. Does this mean FAST will always need to sacrifice the accuracy in change of better fairness levels?

- No discussion in terms of the size of user group ‘B’ and the running time included in the paper. In particular, if we consider a more fine-grained user grouping with less users in each group, will this significantly increase the running time, or have a large impact in terms of the performance?

**Questions:**

See weakness.

**Reviewer Confidence:**

2: The reviewer is willing to defend the evaluation, but it is likely that the reviewer did not understand parts of the paper

**Scope:**

3: The work is somewhat relevant to the Web and to the track, and is of narrow interest to a sub-community

---

### Official Review · Reviewer_Csim · 2023-11-26

**Novelty:** 3
**Technical Quality:** 4

**Review:**

To address the issue of what causes unfairness in meta-learned user modeling, this paper proposes a framework called Fairness-aware Adaptive Sampling for meTa-learning (FAST) which adjusts sampling distribution for user groups during meta-learning's interleaved training. Theoretical guarantees suggest FAST's potential to eliminate unfairness, supported by empirical experiments across three datasets, showing improved fairness without compromising accuracy.

**Questions:**

How to tune the hyperparameters in Appendix A.5?

Some ablation studies are expected.

Any limitations in the proposed method?

**Reviewer Confidence:**

3: The reviewer is confident but not certain that the evaluation is correct

**Scope:**

3: The work is somewhat relevant to the Web and to the track, and is of narrow interest to a sub-community

---

### Decision · Program_Chairs · 2024-01-22

**Decision:**

Accept

**Comment:**

This work focuses on fairness in meta-learning applied to user modeling. Reviewers generally find this to be a compelling study, backed by robust theoretical foundations. They suggest that including additional experiments and analysis could further reinforce the strength of the work.